# Is There a Missing Link? Exploring the Effects of Institutional Pressures on Environmental Performance in the Chinese Construction Industry

**DOI:** 10.3390/ijerph191811787

**Published:** 2022-09-18

**Authors:** Dongmei Lee, Yuxia Fu, Daijiao Zhou, Tao Nie, Zhihong Song

**Affiliations:** 1School of Economics and Management, Shanxi University, Taiyuan 030006, China; 2Institute of Management and Decision, Shanxi University, Taiyuan 030006, China

**Keywords:** institutional pressure, organizational learning, environmental performance, structural equation modeling

## Abstract

Although institutional pressures have huge strategic implications for organizational activities, this certainly does not mean that organizations under institutional pressures can improve environmental performance automatically. Institutional pressures are critical but not sufficient to affect environmental performance directly. Therefore, additional research is needed to explore the missing link between institutional pressures and environmental performance. Based on the “pressure-response-performance” framework, this study integrates perspectives of institutional theory and organizational learning to argue the mediating role of organizational learning in the relationship between institutional pressures and environmental performance. Data were collected via 268 valid questionnaires from construction firms located in Shanxi Province in central China. Hypotheses in the conceptual model were tested with structural equation modeling. Empirical results reveal that both coercive and mimetic pressures have significantly positive effects on organizational learning, whereas normative pressures have a non-significant effect on organizational learning. Besides that, organizational learning has a significantly positive effect on environmental performance. In addition, organizational learning partially mediates the relationship between coercive pressures and environmental performance and completely mediates the relationship between mimetic pressures and environmental performance. By exploring the mediating role of organizational learning, the article uncovers the missing link in the relationship between institutional pressures and environmental performance.

## 1. Introduction

Over the last four decades, China has witnessed unprecedented economic growth while at the same time leading to a precarious ecological situation. Local water and river pollution, regional acid rain, global ozone depletion, and climate change are environmental burdens that negatively affect the sustainability of the Chinese economy. In addition to government regulations, pressures from customers, environmental groups, the media, labor unions, and competitors also pose great challenges for firms. Each of these groups can mobilize public opinion in favor of or against a firm’s environmental practices [1]. For example, many customers require their suppliers to provide them with written certification of their compliance with the environmental regulations. They want that their suppliers manufacture products which do not consist of hazardous materials [2]. Increasing attention and concern over the environmental impact of business has led firms to seek effective ways to minimize their exposure to environmental risk and take substantive measures to reduce the negative effects of environmental burdens. However, others approach the issues more strategically. Green-washing, for example, has been identified by both academia and the mainstream media as a strategy used by businesses to engage in symbolic communications about environmental issues without actually addressing them in substantive actions [3,4]. From the perspective of a rational actor, we might expect firms to act symbolically rather than substantively because appearing to conform is easier and allows for greater internal flexibility than actual conformity while still conferring the benefits of legitimacy [5]. For example, Russo and Harrison [6] discovered that ISO 14001 certification was paradoxically associated with higher toxic air emissions. They speculated that this might be because certification gives a firm the appearance of being green without requiring any substantive actions on its part. As firms’ different actions in response to institutional pressures are closely related to their environmental performance, our first objective is to examine the responses of firms facing institutional pressures in order to fully understand their environmental performance. 

Institutional theory is usually used to explain the impact of institutional pressures on firms’ practices and performance. According to institutional theory, firms are not isolated actors and are constrained by their surrounding environment [7]. They need to take strategic actions to meet the demands of stakeholders in order to gain legitimacy and realize sustainable development. In this way, institutional pressures are isomorphic mechanisms through which institutions influence the behavior of organizations. Previous studies have applied institutional theory to explore the impact of institutional pressures on environmental performance [8,9,10,11,12,13,14,15,16,17,18,19,20]. For example, using firm-level survey data of the leather tanning industry in Mexico, Blackman, and Kildegaard [8] found that environmental regulatory pressures are not associated with pollution reduction. Rather the most significant progress in terms of controlling tannery emissions has resulted from the voluntary adoption of clean tanning technologies. Kassinis and Vafeas [9] empirically investigated the three most polluting industries in the United States: chemicals, primary metals, and public utilities, and found that higher stakeholder pressures are related to lower toxic releases. By collecting data from equipment manufacturing firms in China, Zameer et al. [10] explored the role of green production as a mediator in the relationship between managerial environmental awareness, customer pressures, and regulatory control on environmental performance. Their results indicated that customer pressures and regulatory control are not positively related to environmental performance. The above studies reveal that the impact of institutional pressures on environmental performance remains mixed and controversial. In addition, existing studies on the subject have focused on contexts such as the chemicals, primary metals, and public utilities in the United States [9], the global automotive industry [12], the automotive industry in Spanish [11], and in Brazil [15], manufacturing industry in Pakistan [16] and in China [10,13,17,18,19], leather tanning industry [8] and Agricultural industry in Mexico [20], and electrical and electronics industry in Taiwan province [14], there is still lack of studies on the subject in the context of the construction industry in China. Thus, the second objective of this study is to explore the effect of institutional pressures on environmental performance in a new context of the construction industry in central China.

While the institutional theory has received substantial support in the literature, some recent research has criticized and identified the limitations of the institution-theoretic framework [12,14,21]. Institutional theory is weak in explaining why firms facing the same institutional pressures in the same institutional field have heterogeneous environmental performance [12,20,21]. In addition, the mixed and controversial findings on the relationship between institutional pressures and environmental performance may also suggest that the mechanisms through which institutional pressures drive environmental performance are still largely unclear [18,22]. In other words, there seems to be a “black box” between institutional pressures and environmental performance [22]. How are institutional pressures converted to positive environmental performance? Are there any other factors playing a bridging role in the relationship between institutional pressures and environmental performance? These questions necessitate additional research to explore the missing link between institutional pressures and environmental performance.

Based on Huang et al. [22], the “pressure-response-performance” framework is adopted in this study. Specifically, Following Delmas and Toffel [23], we use “organizational learning” to denote the organizational responses to external institutional pressures. We further distinguish between reactive and proactive organizational learning. While reactive organizational learning is associated with firms’ reactive posture to change and only making small adjustments to the existing environmental practices, proactive organizational learning reflects firms’ proactive posture to change and even aims to switch to completely new forms of environmental practices. It is proposed that institutional pressures can and do lead to the improvement of a firm’s environmental performance in cases where a higher level of organizational learning is present. In other words, it is the different levels (e.g., reactive or proactive) of organizational learning in response to institutional pressures that make the difference in firms’ heterogeneous environmental performance. The reasons are explained below. From the resource-based perspective, an organization learns in order to transform itself and become more adaptable and responsive to external changes and jolts [24,25]. Meyers [25] contended that such environmental shocks provide an opportunity for an organization to learn how to deal with crises. In other words, through organizational learning, an organization improves its performance and readjusts to a changing environment. Firms understand the importance of responding to institutional pressures to help improve their competitive position, which requires them to develop specific capabilities to tackle those pressures [26]. More importantly, in the present era of VUCA (Volatility, Uncertainty, Complexity, and Ambiguity), where organizations are facing challenges such as sustainability, organizational learning plays an especially important role because it serves as a key factor in building organizational capacity and knowledge [27]. A higher level of organizational learning not only provides firms with an advantage in that they can better understand and proactively respond to institutional pressures than their competitors [28] but also provides them with greater capacity to adapt their environmental management practices [18] and therefore achieve higher environmental performance. Previous studies have provided evidence of firms’ changing their practices in response to regulatory pressures but without improving performance [29]. This is particularly the case for firms that take reactive organizational learning in response to institutional pressures and pursue changes only after regulatory pressure forces them to do so [30]. By contrast, firms with a higher level of organizational learning are more inclined to take proactive organizational learning toward environment management beyond just fulfilling the basic regulations and laws on environment protection and thus achieve higher environmental performance [11,30]. Along this line of thought, the article tries to empirically investigate the relationships between institutional pressures, organizational learning, and environmental performance in the context of the construction industry in central China. 

The rest of this article is structured as follows: Section 2 provides the theoretical background and develops research hypotheses. Section 3 describes the data sources, variables measurements, and measurement instrument validation. Section 4 presents the empirical results with SEM. Section 5 discusses the findings and concludes this paper.

## 2. Theoretical Background and Research Hypotheses

### 2.1. Institutional Theory

Over the last three decades, institutional theory has evolved into a potent tool for providing critical insights and rationales for external forces that have a significant impact on organizational actions and decisions [31]. Historically, scholars of institutional theory have focused on how organizations gain institutional legitimacy by adhering to social norms and common organizational practices within the organizational field [7]. As posited by institutional theory, organizational choices are constrained by the invisible pressures of the institutional environment, which plays a binding role in the organization’s pursuit of interest maximization, and organizations obtain stability and legitimacy by satisfying the expectations of the stakeholders [7,32,33]. Organizations must conform to the collective norms and beliefs and must be responsive to external demands and expectations in order to survive [32]. Institutional theorists have identified diverse pressures that delimit and shape organizational action, coercing companies to resemble one another through isomorphism [31]. According to the theory, institutions exert three types of isomorphic pressures on organizations: coercive, mimetic, and normative [7].

Coercive pressures refer to the formal or informal pressures imposed by coercive authorities, such as the government, regulatory agencies, and legal norms. For example, firms may adopt new pollution control technologies to meet the requirements of environmental regulatory agencies. Lack of clear goals or highly uncertain environments (such as demand, supply, or technology) [34,35], the risk of uncertainty may be reduced by learning from the successful experience of other firms. Mimetic pressures stem from the organization’s imitation of the successful practices of competitors or leading firms in the industry. Normative pressures are those composed of social pressures on the organization and its individuals, such as organizational routines, operation processes, professional institutions’ norms and certification, and trade associations. For example, some multinational firms build factories in China and purchase raw materials from local firms. These multinational firms generally require Chinese suppliers to comply with certified environmental management systems, such as ISO 14001 [13]. 

Previous studies have well documented the influence of institutional pressures on environmental performance [15,18,30,35,36]. For example, Simpson [10] found that firms are required by law to publicly report their waste disposal choices, which would presumably encourage firms to actively seek a reduction in waste, thereby leading to better environmental performance. By collecting data from 11 industries, Majid et al. [36] also found a positive relationship between institutional pressures and environmental performance.

Though the institutional theory provides a suitable theoretical framework for understanding the role of institutional pressures in influencing environmental performance, prior research has identified the limitations of the theory [12,16,20,22,30]. Traditionally, institutional theory tends to attribute the persistent heterogeneity among enterprises to the differences in different organizational fields. However, it fails to explain the heterogeneity in firms’ environmental performance even though they are embedded in the same institutional environment. As Hoffman [21] noted, “the form of organizational response is as much a reflection of the institutional pressures that emerge from outside the organization as it is the form of organizational structure and culture that exist inside the organization.” Thus, understanding heterogeneous environmental performance necessitates the consideration of factors of both institutional and organizational characteristics (such as organizational learning, training, and firm resources) [12,22,30].

### 2.2. Organizational Learning Theory

Management literature abounds with definitions of organizational learning. Organizational learning occurs when members of the organization act as learning agents for the organization, responding to changes in the internal and external environment of the organization [37]. According to Bell et al. [38], there exist four disparate schools of thought on organizational learning: an economic school, a developmental school, a process school, and a managerial school. The economic school of organizational learning is concerned with encoding inferences and understanding that result from the repetitive actions and subsequent reactions that occur with continuous production. This approach to learning focuses on the “detection and correction of errors” [37]. The developmental school of thought delivers training via carefully composed and logically structured educational frameworks. Each stage of the education experience represents an incremental development of individual learning and organizational capability [38]. The process school of organizational learning defines it as “a dynamic process of creation, acquisition, and integration of knowledge aimed at developing the resources and capabilities that allow the organization to achieve better performance” [39]. Finally, according to the managerial school, organizational learning occurs as a direct result of a management-led intervention. Managerial intervention is frequently a carefully calculated corporate response to perceived environmental pressures. Because the external environment is constantly changing, a firm must also make continuous increases in its “capacity to realize [its] highest aspirations” [40]. This definition emphasizes a firm’s behavior changes and capacity for action, which is applied in this study. Specifically, considering the context construction industry, we describe organizational learning as the way of construction firms act (reactive vs. proactive) and the extent of changes in practices related to environmental management in response to external environmental pressures. Those environmental management practices include the techniques and practices employed by an organization to tackle environmental challenges, such as building information modeling (BIM), post occupancy evaluation (POE), total quality management (TQM), just-in-time (JIT), and environmental management systems (EMS) [13,14,15,41]. The most intuitive feature of BIM is the 3D visualization in the early stage, which can optimize the engineering design and reduce the possibility of material loss and rework that may exist in the construction stage. The cyclical aspect of continuous improvement of POE, TQM, JIT, and EMS also has as core tenets the concept of waste reduction and elimination, even so far as going to a ”zero waste” goal [13]. Such techniques and policies, aimed at reducing, reusing, and recycling waste, are helpful for construction firms to improve environmental performance. As organizational learning is closely related to (planned) changes in organizations and firms’ responses may be reactive or proactive, the study distinguishes between reactive and proactive organizational learning. While reactive organizational learning entails small adjustments or adaptations of the ways the organizations act (e.g., adjusting existing routines, actions, products, and processes), proactive organizational learning requires far-reaching changes, which can also aim at a switch to completely new forms of satisfying customer needs (e.g., green building technology and process innovations) [42]. An extreme situation of “zero learning” exists when firms are unaware of all of the potential waste reduction options [30] or ignore the implementation of green practices [43]. According to Love et al. [44], while the need for organizational learning is obvious, it is unfortunate that many construction firms are hesitant to embrace TQM as part of their learning to reduce quality failure in their projects. As a result, projects have been plagued by ongoing quality issues and poor environmental performance.

In the relationship between organization and environment, organizations are not only constrained and influenced by the institutional environment but also can proactively respond to the needs of the stakeholders in the organizational field [33]. The higher the environmental pressure is, the greater the need for organizational learning will be [34,35]. Firms that are able to learn to stand a better chance of continuously adjusting to new situations and renewing themselves according to the demands of the environment [45]. As a result, learning organizations are usually more flexible and faster in responding to new challenges than competitors [46].

Based on organizational learning theory, previous literature mainly focused on the effect of organizational learning on firm performance. Jiménez-Jiménez and Cegarra-Navarro [47] examined the relationships between market orientation, organizational learning, and firm performance. Their research showed that market orientation has a positive impact on firm performance only when organizational learning is a mediating variable. By collecting data from 115 firms in mainland China, Song [45] provided additional evidence that organizational learning has a positive effect on innovation. Given few research has examined the role of organizational learning in the relationship between institutional pressures and environmental performance, the paper constructs, and tests the relationships among institutional pressures, organizational learning, and environmental performance in the context of the construction industry in central China.

Considering the limitations of the institutional theory described in Section 2.1, both institutional and organizational characteristics are needed to be jointly considered to understand the heterogeneity in performance [12,21]. An extension of the institutional theory in this study, therefore, is the proposition of organizational learning as a mediator in the relationship between institutional pressures and environmental performance. This point is elaborated as follows. First, organizational learning is essentially about adapting and responding to changes in the institutional environment and is one of the direct approaches for firms to understand and respond to institutional pressures [12,48]. In other words, an organization learns to develop organizational structures and systems to transform itself to become more adaptive and responsive to changes and jolts in the external environment [49]. As Meyer [25] argued, such environmental jolts provide a good opportunity for an organization to learn to deal with external environmental pressures. For example, in response to progressive health and safety legislative requirements, construction firms have begun to learn and apply for relevant certifications, such as environmental management systems and Occupational health and safety management systems (ISO45001). Second, organizational learning is gradually developed in the continuous interaction and coordination of various explorative and exploitative actives. This dynamic capability focuses on continuous adaptation to the changing external environment. In addition, organizational learning is regarded as a means to foster sustainability by institutionalizing new thinking [43]. Thus, “organizational learning” is used in this study to denote the organizational responses to external institutional pressures. It is the different levels of organizational learning which lead to the differences in environmental performance. As such, by linking institutional theory and organizational learning theory, we may provide explanations on questions of why firms facing common institutional pressures have heterogeneous environmental performance [12,16,20]. 

### 2.3. Research Hypotheses

#### 2.3.1. The Influence of Institutional Pressures on Organizational Learning

The institutional theory origins from the idea that firms institutionalize individual and organizational structures by adding either values or constraints to their internal activities or system [31]. According to this theory, organizational decisions or actions are not driven solely by efficiency but by external environmental factors and the need for legitimacy. Compliance with institutional norms may enable firms to obtain various forms of returns, such as obtaining legitimacy and gaining industry recognition [33]. Dodgson [49] emphasized that uncertainty caused by environmental change is the inducement of organizational learning. As the government puts more emphasis on environmental regulation and consumers are increasingly aware of environmental protection, firms must acquire and improve the knowledge related to environmental management through organizational learning. 

As discussed earlier, there are three dimensions of institutional pressures: coercive pressures, mimetic pressures, and normative pressures. Coercive pressures stem from mandatory regulatory policies imposed by government authorities. Environmental protection laws and regulations by central and local governments are considered to be important forces affecting the energy conservation and emission reduction practices adopted by Chinese firms. Up to now, the Chinese government has issued a number of environmental protection laws and regulations to reduce resource consumption and pollution emissions. For example, the environmental protection tax law of the People’s Republic of China, issued in December 2016, may prompt firms to internalize the negative externalities caused by pollution in the production processes and take action to reduce the level of pollution emissions.

Previous studies have found that coercive pressures have a positive impact on firms’ organizational learning related to environmental management practices [13]. As a mandatory authority, the government plays an important role in firms’ organizational learning and adoption of environmental management practices [11,12,13,16,20,22,30]. On the one hand, the government explicitly supports firms that adopt environmental management practices by providing subsidies, tax relief, or other financial incentives to firms [16]. On the other hand, firms must comply with government regulations or face the threat of regulators levying legal action, penalties, and fines [11,13,30]. Either way, firms must learn how to adapt their organizational practices to accommodate coercive government pressures. Zhu et al. [17] investigated the four manufacturing industries of chemical, electronic, automobile, and machinery in the coastal areas of eastern China. They found that coercive pressures exerted by the Chinese government have a significant positive impact on firms’ environmentally-oriented organizational learning. Using a sample of 259 Chinese manufacturing firms, Zhang and Zhu [18] argued that firms, especially those in emerging economies such as China, are more likely to respond to regulation pressures by reactive organizational learning (i.e., making small adjustments to existing environmental practices). 

As regards the construction industry, China did not have any unified construction law until 1996. The Construction Law was enacted on 1 November 1997. According to the Construction Law, construction firms should abide by the provisions of the laws and regulations on environmental protection and production safety and take measures to control and deal with all kinds of dust, exhaust gas, wastewater, solid waste and noise, vibration pollution that are harmful to the environment at the construction site. Construction firms’ failing to comply with government regulations would ultimately result in legal action, penalties, and fines [22]. For rational actors in the construction industry, it is plausible to learn environmental management practices to avoid being punished. Therefore, the hypothesized relationship between coercive pressures and organizational learning is stated as follows:

**Hypothesis** **1a** **(H1a).**
*Firms facing a higher level of coercive pressures positively impact organizational learning.*


When organizational technologies are poorly understood, when goals are ambiguous, or when the environment creates symbolic uncertainty, an organization may model itself on other organizations [7,34,35,49]. Mimetic pressures result in organizational efforts to copy or imitate the other firms who are performing best in the market or, in other words, to follow the benchmark [36]. For instance, a firm’s inclination toward exploitation or exploration learning “may be influenced by institutional pressures such as the search behavior of its competitors or of those in its social network” [50]. Wong et al. [51] argued that imitations of the best practices in the industry might help to improve the competitiveness of firms and avoid environmental uncertainty in the operation process. Organizational learning is a complex, multifaceted phenomenon. According to Bergh [52], there are two types of organizational learning: the direct experience of trial-and-error experimentation and the indirect experience of imitating positive and negative aspects of other firms through observation. As the knowledge base of focal firms and their competitors is highly related, imitation of competitors is an easy way to gain their positive or negative experiences. With the increased globalization, Chinese firms have gained more opportunities to learn from their counterparts. For example, Chinese firms (especially firms in the field of energy-saving buildings) are gradually imitating their competitors, striving to introduce environmental management systems and implement the ISO 14001 environmental management policy [13]. In other words, when firms are faced with greater mimetic pressures, firms will proactively learn and benchmark the environmental management practices of the leading firms in the industry in order to gain a competitive position. 

In the construction industry in China, foreign competitors are bringing in different and innovative systems and approaches to construction. For example, Building Information Modeling (BIM) technology, originating in Europe and the USA, has brought extraordinary changes to the traditional construction industry [53]. BIM is a multifaceted computer software data model that applies virtual and augmented reality technologies to visually present an architectural design while integrating capital and recapitalized installation into the design based on user requirements and feedback [41]. As a technological innovation, the emergence of BIM allows relevant construction firms to quickly and accurately obtain basic engineering data, providing effective support for construction enterprises to develop accurate workforce plans, greatly reducing waste in resources, logistics, and storage links, and providing technical support for the realization of consumption control. Recognizing the business and operational advantages of BIM technology that their foreign counterparts have gained, Chinese construction firms are gradually interested in learning how to apply BIM technology in construction projects [41,53]. Therefore, the hypothesized relationship between mimetic pressures and organizational learning is stated as follows:

**Hypothesis** **1b** **(H1b).**
*Firms facing a higher level of mimetic pressure positively impact organizational learning.*


Normative pressures stem primarily from professionalization and “generally take the form of rules-of-thumb, standard operating procedures, occupational standards, a deductional curricula” [54]. Normative pressures affect organizational learning in two ways. The first is the product standards and the environmental quality certification system adopted in the industry. When the production processes or technologies adopted by firms do not conform to the environmental standards set by the industry association, firms may lose the opportunity to enter the specific market. The second is the demand by customers. Customers’ requirements form a core normative pressure [11,13] and are cited as an important driving force in inducing firms to take environmental initiatives [19]. Christmann and Taylor [55] found that customers from developed countries required Chinese firms to improve environmental management practices and adopt the ISO14001 environmental management system. With the increased worldwide pollution prevention and protection awareness by the public, Chinese firms are also aware of the importance of building an environment-friendly image, which may prompt firms to proactively search for information and develop new knowledge of products, processes, and services related to environmental management practices [13]. 

In the context of the construction industry, construction clients, especially those with significant capital expenditure, can exert substantial influence on their commercial relationships with first-tier construction service and product providers [56]. Consequently, the role of the construction client in construction firms has become increasingly distinctive. Nowadays, construction clients have high expectations of construction buildings. They expect their buildings to be energy efficient, environmentally friendly, and not impact their health negatively [57]. In order to satisfy construction clients’ demands and capture market share, construction firms have to proactively respond to their pressures and learn and utilize innovative construction technologies and management practices (e.g., BIM, POE, EMS, and TQM). Therefore, the hypothesized relationship between normative pressures and organizational learning is stated as follows:

**Hypothesis** **1c** **(H1c).**
*Firms facing a higher level of normative pressure positively impact organizational learning.*


#### 2.3.2. The Influence of Institutional Pressures on Environmental Performance

Environmental performance is generally defined as firms’ performance to “minimize or eliminate emissions, effluents, and waste from their operations” [58]. In recent years, the Chinese government has enacted a number of environmental regulations due to the increasing problems of dwindling resources and environmental pollution. The Law of the People’s Republic of China on Prevention and Control of Air Pollution promulgated in 2015 requires enterprises to take effective measures to prevent and reduce air pollution and to take responsibility for the damage caused by pollution. From 2015 to 2016, the State Council implemented nationwide strategies to prevent and control water, air, and soil pollution, requiring companies to reduce energy consumption and pollutant emissions. In addition, environmental regulations issued by the Chinese environmental protection authorities are becoming increasingly stringent. Trade barriers and administrative penalties are common methods of punishment. These regulatory policies, derived from coercive powers, require companies to adopt environmental management measures to reduce their negative impact on the environment. 

Firms must comply with environmental regulations or face penalties, fines, class action lawsuits, and, in the worst-case scenario, removal from the market [11,13,30]. Simpson [30] discovered that domestic regulations and disposal cost pressures were significant drivers of environmental performance in a sample of US-based manufacturers. He contended that businesses are required by law to publicly report their waste disposal choices, which would presumably encourage businesses to seek a reduction in either total reported waste or waste sent to landfill. By using data from the agricultural sector in Mexico, Juárez-Luis et al. [20] also found that coercive pressures directly affect farmers’ environmental conservation. In addition, governments of the UK, Australia, Singapore, Sweden, and Hong Kong SAR have implemented policy mandates, making BIM use for the delivery and management of public projects compulsory [59]. Firms facing coercive pressures to adopt BIM technology can effectively improve efficiency, reduce costs, and save resources. This leads to the following hypothesis:

**Hypothesis** **2a** **(H2a).**
*Firms facing a higher level of coercive pressure are more likely to achieve better environmental performance.*


Bergh [52] argued that firms pay close attention to other firms, such as their competitors, and are more likely to be influenced by pressures from competitors than by legal or regulatory requirements. Thus, environmental performance may also be affected by mimetic pressures. In addition, firms can learn positive environmental management practices from leaders in their industry and gain inspiration from the environmental management practices of their competitors. By generating an incentive to imitate among competitors based on voluntary choice, mimetic pressures, therefore, have the potential to influence environmental performance [60]. According to institutional theory, enterprises often imitate the structure and behavior of other organizations in the same environment to meet the requirements of legitimacy so as to alleviate external risks [61]. Juárez-Luis et al. [20] also argued that small agricultural businesses in Mexico tend to imitate the green practices of the competitors they consider successful or those closest to their facilities, which enable them to improve their environmental performance. 

Prior studies have found that pressure from competitors is a driving force behind the adoption of information modeling and simulation technologies in the construction industry [41]. Under intense competition, a firm is more likely to seek out novel ways of doing business and, as a result, imitate other firms. This is common in Chinese construction firms. After seeing the relative benefits of environmental management practices, firms in the construction industry are more likely to learn these practices. Since 2013, China State Construction has led the industry to carry out the construction of BIM demonstration projects, and more construction firms are starting to imitate the practices, resulting in improved environmental performance [41,53]. This leads to the following hypothesis:

**Hypothesis** **2b** **(H2b).**
*Firms facing a higher level of mimetic pressure are more likely to achieve better environmental performance.*


When firms feel pressures from external standardization, voluntary norms, and professional practice, there is an urgent need for managers to rigorously and passionately follow external technical guidance and organizational models [31]. Thus, normative pressures may induce firms to internalize environmental management systems, promote deeper and more widespread implementation of environmental management practices, and positively influence managers’ efforts to invest economic resources with the aim of improving environmental performance through planning, developing, and implementing innovations [60].

In addition, capital markets may show opposing attitudes to the disclosure of positive or negative environmental events (such as permit violations, waste pollution, court proceedings, and customer complaints). Disclosure of negative environmental events by some companies may damage their reputation across the industry, and industry associations may use professional means to encourage companies to focus on and improve their environmental performance so that the public and government do not lose confidence. Environmental concerns of suppliers and customers also require firms to provide evidence of improved environmental performance. Through immediate responses to customer pressures, firms can build favorable and diverse networks with their customer and pick up novel and innovative ideas to help them develop green product innovation [1,18]. Construction firms that fail to enhance their performance and meet their customers’ needs and achieve their satisfaction encounter the risk of losing customers, competitiveness, and market share. Upstill-Goddard et al. [46] used the case study approach and abductive logic to understand what role learning plays with regard to sustainability standard implementation. Their findings indicated that customer demand was a key reason for construction firms to initiate the BES 6001 framework standard implementation project. If construction firms want to present a “green” image of sustainable consumption to customers, they need to improve their environmental performance to achieve this goal. All of these lead to the following hypothesis, 

**Hypothesis** **2c** **(H2c).**
*Firms facing a higher level of normative pressure are more likely to achieve better environmental performance.*


#### 2.3.3. The Influence of Organizational Learning on Environmental Performance

In a changing environment, firms need to encode and store the knowledge of their employees and then translate it into shared knowledge for the members of the organization through organizational learning, including rules, beliefs, procedures, and routines [62]. March [63] stated that in adapting to the external environment, firms must balance exploitative learning (leveraging existing knowledge) and exploratory learning (pursuing new knowledge). Exploitative learning refers to reducing production costs and increasing resource efficiency and productivity by improving existing production techniques and process technologies, while exploratory learning is characterized by the discovery and exploration of new knowledge, such as the development of new process technologies to prevent and control pollution. As we have discussed in Section 2.2, firms may take reactive or proactive organizational learning in response to institutional pressures. When firms take reactive organizational learning, they tend to be conservative and slow to change the existing environmental practices, which may contribute little to improvement in environmental performance. Instead, when firms take proactive organizational learning, they usually aim to switch to a completely new way of acting, which can contribute to environmental performance. Previous studies have shown that organizational learning positively influences firm performance [22,45]. For example, Huang et al. [22] provided evidence of the positive effect of training (a form of organizational learning) on green innovation performance. 

In the context of the construction industry, firms adopting and learning construction technologies and environmental management practices, such as BIM, POE, TQM, JIT, and EMS, would improve their environmental performance. The reasons are explained as follows. First, the BIM technology adopted by construction firms can be used to analyze all aspects of building performance, including lighting, energy efficiency, and sustainable materials that affect green conditions. In addition, it can analyze, achieve the lowest energy consumption, and realize energy saving and environmental protection with the help of ventilation, lighting, and airflow organization. Second, “Lessons Learned” is a common concept in the practice of designing and constructing buildings [57]. Typically, when a building is built, the project is revisited after completion to find out what worked and what did not work, and the lessons are applied by continuing what worked and avoiding what did not. The lessons should change continually, as old lessons are incorporated into an ever-improving professional practice, and new, more nuanced lessons are identified and added. Through such learning iteration and continuous improvements, firms may improve their environmental performance. Finally, the cyclical aspect of continuous improvement of TQM, JIT, and EMS also has as core tenets the concept of waste reduction and elimination, even so far as going to a “zero waste” goal [13]. Such techniques and policies, aimed at reducing, reusing, and recycling waste, are helpful for firms to improve environmental performance. Based on the above reasoning, this article puts forward the following hypothesis:

**Hypothesis** **3** **(H3).**
*Organizational learning positively affects firms’ environmental performance.*


#### 2.3.4. The Mediation role of Organizational Learning

Hypotheses 1a, Hypotheses 1b, Hypotheses 1c, and Hypotheses 3 link institutional pressures and environmental performance with organizational learning. If the hypotheses are tested as a set, it suggests a mediating role of organizational learning in the relationship between institutional pressures and environmental performance. In other words, institutional pressures may indirectly influence environmental performance through organizational learning. The elaboration of this point is as follows.

As stated in Section 1, institutional theory identifies three kinds of institutional pressures (coercive, normative, and mimetic) which create isomorphism in organizational strategies, structures, and processes [64]. Some recent studies have criticized this isomorphism [12,16,22,65], noting the heterogeneity in firms’ response to institutional pressures. One reason may be that each firm interprets institutional discourses and practices through its own lens [12]. As Gilbert [66] noted, external pressures do not typically trigger an organizational response until amply perceived and understood. It is, therefore, necessary and important to consider the mediation variable in order to understand the heterogeneity when firms respond to external pressures [12,22]. This study considers organizational learning as a mediator. Organizational learning provides firms with knowledge and experience in response to institutional pressures [48], which signifies the ability of firms’ to make ongoing adjustments in resource allocation and build new organizational thought [66]. A higher level of organizational learning not only provides firms with a better understanding of institutional pressures and greater capacity for action [67] but also buffers firms from uncertainty and enables them to take risky initiatives in response to external pressures. Firms with a higher level of organizational learning are better equipped with knowledge and capacity to adopt environmental management practices in an effort to minimize environmental burdens through the proactive development of innovative solutions. By contrast, Firms with a lower level of organizational learning are inherently reactive and conservative and are endowed with less knowledge and experience in managing institutional pressures. They are inclined to adopt quick, simple solutions, which may contribute little to improving environmental performance [30].

Coercive pressures may come in the form of a command-and-control mechanism and an incentive and voluntary mechanism [20,68]. The command-and-control mechanism allows regulatory agencies to impose restrictions through authoritative orders, and firms will suffer penalties for noncompliance. Incentive and voluntary mechanisms allow businesses to obtain subsidies or other concessions. In response to the coercive pressures, construction firms—especially those in developing economies such as China—are more inclined to pursue reactive organizational learning by adapting the existing production processes [18]. In this case, firms can achieve only short-term improvement in environmental performance since the existing production processes may suffer from technological obsolescence [12]. Using a sample of U.S.-based manufacturers, Simpson [30] found that firms facing international and domestic regulations may not produce beneficial performance outcomes where firms lack the relevant knowledge and capacity to understand and convert ideas into action at the plant level. 

Under mimetic pressures, firms may pay more attention to learning and imitating the behavior of competitors in the same industry in order to avoid uncertainty and maintain their social image [25,49,69]. More importantly, in response to the increasingly strict mimetic pressures, as firms competing in the industry learn how to overcome specific challenges, they develop valuable knowledge and capabilities, which they can leverage to take proactive postures and adopt more advanced methods in an effort to minimize environmental burdens through the development of clean products and technologies [12]. As Ye et al. [19] have noted, as multinational companies (MNCs) invest in China, Chinese firms faced with new foreign competition are under increasing pressure to implement environmental protection practices. In addition, according to the ‘competition leads to competence’ approach [70], mimetic pressures pose opportunities and threats for firms. As construction firms imitate the environmental practices (e.g., assembled building, BIM) of the leading firms in the industry, they may develop potentially valuable resources and capabilities, which in turn can provide firms with better environmental performance.

Normative pressures arise from social obligation or professionalization and have been found to encourage firms to adopt environmental management practices in order to gain legitimacy. Customers’ requirements form a core normative pressure and have proven to be an important driver for construction firms to improve environmental performance. As the level of environmental consciousness across society has risen, customers may have developed a preference for eco-friendly constructions, which may push firms to search for innovative solutions and thus improve environmental performance [18]. In order to respond to customers’ demands, Othman and Elsaay [57] proposed a learning-based framework by adopting POE, a learning tool that helps to learn through previous feedback from previous construction projects, thus contributing to improving the environmental performance of the new construction projects. 

Thus, this article puts forward the following hypothesis:

**Hypothesis** **4a** **(H4a).**
*Organizational learning plays a mediating effect in the relationship between coercive pressures and environmental performance.*


**Hypothesis** **4b** **(H4b).**
*Organizational learning plays a mediating effect in the relationship between mimetic pressures and environmental performance.*


**Hypothesis** **4c** **(H4c).**
*Organizational learning plays a mediating effect in the relationship between normative pressures and environmental performance.*


The above hypotheses can be illustrated in Figure 1.

## 3. Research Method

### 3.1. Study Setting and Data Collection

The construction industry in Shanxi Province in central China was chosen as the context for this study for several reasons. Firstly, the construction industry is not only one of the most polluting industries in terms of consuming disposable resources and energy, but it is also a significant source of air pollution, water pollution, greenhouse effect, and solid waste pollution. Secondly, as ecological protection, green development, and low-carbon development have become the focus of social attention, Chinese construction firms are increasingly facing pressures from various stakeholders, including environmental standards set by government regulators for energy consumption and pollution control limits, environmental management practices adopted by competitors, and incentives from users, suppliers, and industry associations to save energy and protect the environment. Thirdly, Shanxi Province often referred to as a heavily polluted region, has a high level of pressures and environmental concern from government regulators, providing an ideal context for testing the relationships between institutional pressures, organizational learning, and environmental performance. 

With the assistance of local government agencies, 300 construction firms in Shanxi Province were randomly selected from a list provided by the government agencies. The data were obtained through the key informant technique [71]. The survey respondents were project managers of construction companies. These managers had many years of experience in the construction industry and were supposedly knowledgeable about the issues under study, which would ensure a considerable degree of external validity of the findings. 

Questionnaires were then sent out to managers along with a covering letter introducing the study as well as the strict confidential commitment. The survey lasted 3 months. A total of 300 questionnaires were sent out, and 268 valid questionnaires were collected. The average response rate was 89.3%. In this paper, a T-test was used to compare the differences between the respondents and non-respondents in the characteristics of sample firms and model variables. The results show that there is no significant difference, indicating there was no non-response bias.

### 3.2. Measures

Measurements of the constructs were adapted from or developed on the basis of prior research (see Appendix A). For each latent variable, at least three items are used for measurements. The five-point Likert scale was used in all the items in the questionnaire, ranging from “1 = disagree strongly” to “5 = strongly agree”.

Environmental performance (ENVPER) was measured by a 6-item scale of an adapted version of measures developed by Vanalle et al. [15], Simpson [30], and Dubey et al. [34]. The environmental performance scale reflects subjective ratings of results on environmental management in terms of raw material use, energy conservation, and waste reduction of construction firms.

Coercive pressures (COEPRE) were measured by a 4-item scale of an adapted version of measures developed by Juárez-Luis et al. [20], Gunarathne et al. [32], and Dubey et al. [34]. Coercive pressures reflect demands from government regulators such as pollution control, reduction of energy consumption, and reduction in the use of toxic/hazardous/hazardous materials.

Mimetic pressures (MIMPRE) were measured by a 4-item scale of an adapted version of measures developed by Juárez-Luis et al. [20], Dubey et al. [34], and Daddi et al. [60], which captures environmental protection pressures from leading companies or competitors in the construction industry. 

Normative pressures (NORPRE) stemming from customers and industry associations were measured by a 4-item scale of an adapted version of measures developed by Juárez-Luis et al. [20], Daddi et al. [60], which captures the demands and expectations from customers and industry associations on environmental protection.

Organizational learning (ORGLER) was measured by an 8-item scale of an adapted version of measures developed by Tu and Wu [27], Cui and Wang [48], and Bhatia and Jakhar [69]. This study measures organizational learning as a single construct, capturing the way construction firms act (reactive vs. proactive) and the extent of changes in practices related to environmental management in response to external environmental pressures [12,15].

## 4. Data Analysis and Results

### 4.1. Descriptive Statistics

Means, standard deviations, and correlations between constructs are presented in Table 1.

### 4.2. Measure Validation

A series of tests were conducted to assess the reliability and validity of the constructs before testing the hypotheses. Cronbach’s α and composite reliability were used as measures of scale reliability [72]. Table 2 shows that Cronbach’s α and composite reliability of all scales were greater than the threshold value of 0.7, indicating high reliability of the scales [73].

Secondly, the article used confirmatory factor analysis (CFA) to test for convergent validity. The results of the confirmatory factor analysis are provided in Table 2. As shown in Table 2, all factor loadings were statistically significant, indicating evidence of convergent validity. In addition, the average extracted variance (AVE) for each construct was greater than the threshold of 0.5, providing further evidence of convergent validity [74].

Finally, discriminant validity was assessed by comparing the AVE of each latent construct with the square of the correlation coefficient between the focal construct and each of the other constructs (see Table 1 and Table 2). Since the AVE of each latent construct was greater than the square of the correlation coefficient between the focal construct and each of the other constructs, providing evidence of discriminant validity.

As our data were all obtained from the same source, we performed Harman’s one-factor test to check for the common method bias. The factor analysis showed that all variables yielded a factor solution, accounting for 62.877% of the total variance, with the first factor accounting for only 19.326% of the total variance. The absence of a single dominant factor and the fact that the first factor did not explain the majority of the variance suggests that common method variance was not a problem in this study.

### 4.3. Structural Model

Structural equation modeling (SEM) with a maximum likelihood estimation procedure was used to test the hypotheses. Table 3 shows the model fit indices, indicating high goodness of fit between the model and the data.

Figure 2 shows the results of the parameter estimation. Table 4 summarizes the results of hypothesis testing. The path coefficient from coercive pressures to organizational learning is 0.185 (t = 2.791, *p <* 0.01), suggesting that coercive pressures have a positive impact on organizational learning. The path coefficient from mimetic pressures to organizational learning is 0.483 (t = 6.327, *p <* 0.01), suggesting that mimetic pressures have a positive impact on organizational learning. The path coefficient from normative pressures to organizational learning is −0.056 (t = −0.892, *p* > 0.1), indicating that normative pressures have no significant impact on organizational learning. Therefore, H1a and H1b are supported, while H1c is not supported. In addition, the path coefficient from coercive pressures to environmental performance is 0.261 (t = 4.134, *p <* 0.01), suggesting coercive pressures have a positive impact on environmental performance. Hence H2a is supported. The path coefficient from mimetic pressures to environmental performance is −0.054 (t = −0.761, *p* > 0.1), indicating that H2b is not supported. The path coefficient from normative pressures to environmental performance is 0.045 (t = 0.788, *p* > 0.1), indicating that H2c is also not supported. The path coefficient from organizational learning to environmental performance is 0.609 (t = 6.976, *p <* 0.01), suggesting that organizational learning has a positive impact on environmental performance. Therefore, H3 is supported.

### 4.4. Testing the Mediation Effects

The mediating effect of organizational learning in the relationship between institutional pressures and environmental performance was tested by the bootstrapping method [75]. By drawing a bootstrap sample of 5000, we test the hypotheses with a 95% confidence interval. The results of the mediating effect test are shown in Table 5. As can be seen from Table 5, the indirect effect of coercive pressures on environmental performance was 0.113 with a 95% confidence interval of [0.028, 0.189], which does not include the value of 0, indicating that organizational learning mediates the relationship between coercive pressures and environmental performance. Furthermore, the direct effect was estimated at 0.261 with a 95% confidence interval of [0.128, 0.385], excluding the value of 0, indicating that organizational learning partially mediated the relationship between coercive pressures and environmental performance. Therefore, H4a is supported.

The indirect effect of mimetic pressures on environmental performance was 0.294 with a 95% confidence interval of [0.199, 0.413], which does not include the value of 0, indicating that organizational learning mediates the relationship between mimetic pressures and environmental performance. However, the direct effect was estimated at −0.054 with a 95% confidence interval of [−0.208, 0.105], which does include the value of 0, indicating organizational learning plays a completely mediating effect in the relationship between mimetic pressures and environmental performance. Hence, H4b is also supported.

The indirect effect of normative pressures on environmental performance was −0.034 with a 95% confidence interval of [−0.115, 0.041], which does include the value of 0, suggesting that organizational learning does not play a mediating role in the relationship between coercive pressures and environmental performance. Hence, H4c is not supported.

## 5. Discussions and Implications

### 5.1. Summary of the Findings 

Although institutional pressures have huge strategic implications for organizational activities, this certainly does not mean that organizations under institutional pressures can improve environmental performance automatically. Institutional pressures are critical but not sufficient to affect environmental performance directly [16]. Therefore, additional research is needed to explore the missing link between institutional pressures and environmental performance. By linking institutional theory and organizational learning theory, the article investigates the relationships between institutional pressures and environmental performance, as well as the mediation role of organizational learning. Empirical studies show that coercive pressures have a significant positive impact on organizational learning, implying that coercive pressures exerted by central and local governments is an important factor in motivating firms to adopt environmental management practices. This corroborates a series of studies on the topic [13,20]. Coercive pressures are generally considered as key determinants of environmental performance [13,16,20,69]. Coercive pressures may require businesses to reduce energy consumption, control emissions of pollutants, and reduce the use of toxic/hazardous/hazardous materials. Such pressures may induce companies to adopt environmentally-related organizational learning practices to avoid penalties for failing to meet government expectations [76]. In addition, the empirical results show that mimetic pressures have a significant positive impact on organizational learning. When companies perceive that their competitors have adopted environmental management practices (e.g., ISO14001 environmental management systems), they are also willing to take appropriate catch-up measures, such as using environmental standards to improve their environmental performance. After China’s entry into the WTO, Chinese firms are increasingly facing competitive pressures from their foreign counterparts, especially those multinational companies operating in the Chinese market [13]. Our finding is consistent with previous literature [13,16]. For example, Saeed et al. [16] examined whether different kinds of institutional pressures affect external green supply chain management (GSCM) practices (e.g., green purchasing, cooperation with customers for the environment, and investment recovery) in the manufacturing industry in Pakistan. They found that mimetic pressures positively influence external GSCM practices. Zhu et al. [13] also found that international mimetic pressures (i.e., green strategies by foreign producers of similar products) are also positively related to total quality environmental management practices, indicating that foreign competitors in China have motivated Chinese manufacturing firms to implement environmental management practices, which may lead to improve environmental performance. 

Our findings confirm that organizational learning mediates the relationship between institutional pressures and environmental performance. Specifically, organizational learning partially mediates the relationship between coercive pressures and environmental performance, implying that coercive pressures not only have a direct impact on environmental performance, but also have an indirect impact on environmental performance through organizational learning. In contrast, organizational learning plays a completely mediating role in the relationship between mimetic pressures and environmental performance, suggesting that imitation pressures have no direct effect on environmental performance, but only have an indirect effect on environmental performance through organizational learning. As new technologies, materials, and techniques are constantly emerging in the Chinese construction industry, the use of assembled buildings, BIM, steel structures, self-compacting concrete, and new insulation materials have stimulated construction companies to acquire new knowledge related to environmental management through organizational learning and convert this knowledge into environmental management practices, thus contributing to improved environmental performance.

Although previous studies have documented evidence of the positive effect of normative pressures on environmental performance [17], we find no support for the relationship. Zhu et al. [17] claimed that normative pressures positively affect environmental performance. Their argument is based on the fact that China has increasingly attracted foreign investments since the opening-up policy, providing additional motivation for Chinese manufacturers to serve foreign customers by improving environmental performance. Zhang et al. [35] showed that customer and supplier demands for energy efficiency and environmental protection are important sources of normative pressure for Chinese firms. Especially after China’s entry into the WTO, Chinese firms have to comply with the strict environmental standards and low-carbon requirements of their foreign partners in order to remain competitive in international business. We offer two possible explanations for this finding. Firstly, despite the increased level of environmental awareness across society, customers in the Chinese construction industry may be in a weak position to demand green and eco-friendly products. Because pressures from customers are not a high priority for Chinese construction companies, the reactive and conservative posture of construction firms may lead to minor changes in existing environmental practices, which contributes little to improving environmental performance. Secondly, for the construction industry in China, an environmentally friendly social climate, which may inspire the adoption of socially responsible environmental practices to protect the environment and conserve energy, has not yet been developed by the public. Thus, the impact of normative pressure may be ignored by construction firms. This point echoes Seles et al. [77] who proposed that normative pressures tend to be more effective than coercive pressures in the context of an environmentally mature sector, such as the car battery sector. In other words, the construction industry in central China in this study cannot be considered to have sufficient maturity, which may possibly explain the poor results concerning environmental performance. 

### 5.2. Theoretical Implications

This study contributes to the literature in three ways. First, the objective of this paper is to examine the impact of three dimensions of institutional pressures on environmental performance in the context of the construction industry in central China. Previous studies have examined the impact of institutional pressures on environmental performance in sectors such as the leather tanning industry [8], automotive industry [11,12,15], manufacturing industry [10,13,16,17,18,19], electrical and electronics industry [14], and agricultural industry [20]. However, there is still a lack of studies that analyze the subject in the context of the construction industry in China. Thus, this paper contributes to the existing literature by investigating the impact of institutional pressures on environmental performance in a context not previously studied. As Edmondson and McManus [78] have pointed out, when a mature theory that presents well-developed constructs and models that have been studied over time (such as institutional theory) is used as a theoretical background, studies contribute to the field of research by testing the theory in a new setting. 

Second, although previous studies have confirmed that institutional pressures may have important strategic implications for organizational activities [7,16,34,35], this certainly does not mean that organizations under institutional pressure can improve environmental performance automatically. Institutional pressures are critical but may not be sufficient to affect environmental performance directly [36]. In addition, some recent studies have criticized the institutional theory [12,16,20,21], which fails to explain the heterogeneity in firms’ environmental performance when facing the same institutional environment. Theoretically, this paper integrates perspectives of institutional theory and organizational learning to argue the mediating role of organizational learning in the relationship between institutional pressures and environmental performance. Our findings contribute to institutional theory by echoing Ingenbleek and Dentoni’s [79] call to integrate organizational learning theory in order to explain the mechanism through which institutional pressures influence environmental performance. Previous studies have explored other mediators in the relationship between institutional pressures and environmental performance, such as training [11], organizational ambidexterity [12], green supply chain management (GSCM) [15], environmental concern [18], green organizational responses [22], knowledge resources [30], and comprehensiveness of environmental management systems [70]. In summary, by uncovering the mediating mechanisms of organizational learning, we complement prior studies.

Finally, the direct relationship between institutional pressures and environmental performance is rarely explored in existing literature [36]. However, some evidence is available on the indirect link between institutional pressures and environmental performance (See, e.g., [12,15,18,22,30]. Our findings suggest that coercive pressures directly affect firms’ environmental performance, which provides further evidence of the effect of institutional pressures on environmental performance in the context of the construction industry in China. Therefore, our study contributes to the previous literature on the relationship between institutional pressures and environmental performance.

### 5.3. Practical Implications

This study also offers some important practical insights. Firstly, policymakers should be fully aware of the positive impact of government regulation and policy orientation on the environmental performance of enterprises. Thus, central and local government regulators need to develop and improve a series of mandatory environment-friendly policies, such as environmental laws, regulations, and standards, to control the polluting emission behavior of firms. Secondly, policymakers should create an orderly competitive environment for construction firms. Promoting successful environmental management practices as benchmarks in the industry will help construction firms to better learn from leading competitors and thus improve environmental performance. Finally, as organizational learning partially mediates the relationship between coercive pressures and environmental performance, another way of improving environmental performance may be creating market opportunities for construction firms to learn environment-friendly management practices, such as green construction demonstration projects. Policymakers should implement preferential policies or economic incentives, such as low-interest loans and financial subsidies, to guide construction firms to voluntarily and proactively adopt environmental management practices that go beyond legal compliance. In other words, policymakers should give equal weight to coercive pressures and policy incentives in order to achieve the goal of green development effectively.

### 5.4. Limitations and Directions for Future Research

This study suffers from some limitations that future research should overcome. First of all, the article takes the construction industry in central China as the research context. On the one hand, although there are some common features in the eastern, western, and central regions of China, these regions still exist in heterogeneity (such as education and economic conditions). On the other hand, although a homogenous sample from the construction industry may provide deeper insights into the relationships among institutional pressures, organizational learning, and environmental performance, the conclusions in this study may not be generalizable to other industries. Testing other industries in different regions of China is one of the directions in further research. Second, it is difficult to establish a causal relationship between institutional pressures, organizational learning, and environmental performance with cross-section data. Future research may use longitudinal data to overcome the limitations of the study.

## Figures and Tables

**Figure 1 ijerph-19-11787-f001:**
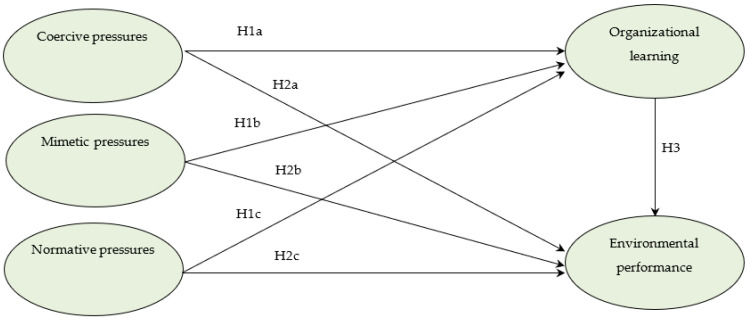
Conceptual Model.

**Figure 2 ijerph-19-11787-f002:**
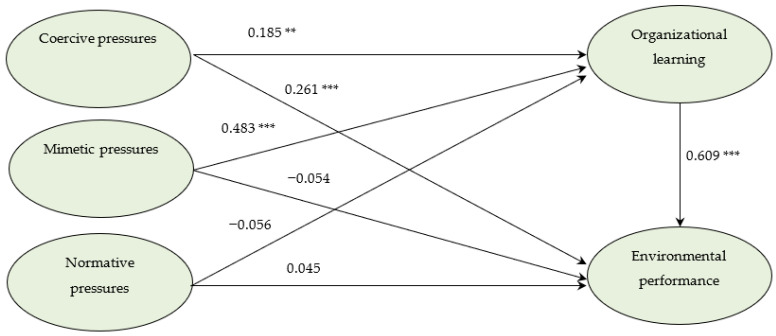
Structure Model. Note: *** *p <* 0.01, ** *p <* 0.05.

**Table 1 ijerph-19-11787-t001:** Descriptive Statistics and Construct Correlations.

Construct	Mean	Standard Deviation	CORPRE	MIMPRE	NORPRE	ORGLER	ENVPER
CORPRE	4.2285	0.45567	0.527				
MIMPRE	3.8797	0.50301	0.146 *	0.508			
NORPRE	3.6091	0.70444	0.112	0.141 *	0.549		
ORGLER	4.2094	0.46336	0.226 **	0.443 **	0.043	0.517	
ENVPER	4.1959	0.45828	0.351 **	0.247 **	0.063	0.566 **	0.535

Notes: Diagonal elements are average variance extracted (AVE) values; Off-diagonal elements are the correlations of the constructs. ** *p <* 0.05, * *p <* 0.1.

**Table 2 ijerph-19-11787-t002:** Measurement Reliability and Validity.

Construct	Items	Standardized Factor Loadings	t-Value	Cronbach’s α	Composite Reliability
Coercive pressures	CP1	0.794 ^a^	--	0.812	0.728
	CP2	0.813	12.263 ***		
	CP3	0.616	9.199 ***		
	CP4	0.661	10.491 ***		
Mimetic pressures	MP1	0.778 ^a^	--	0.792	0.780
	MP2	0.576	8.792 ***		
	MP3	0.840	12.065 ***		
	MP4	0.625	9.642 ***		
Normative pressures	NP1	0.841 ^a^	--	0.821	0.783
	NP2	0.839	14.328 ***		
	NP3	0.692	11.39 ***		
	NP4	0.554	8.678 ***		
Organizational learning	OL1	0.682 ^a^	--	0.893	0.797
	OL2	0.590	8.918 ***		
	OL3	0.804	11.759 ***		
	OL4	0.710	10.492 ***		
	OL5	0.729	10.784 ***		
	OL6	0.772	11.287 ***		
	OL7	0.758	11.244 ***		
	OL8	0.685	10.232 ***		
Environmental performance	EP1	0.721 ^a^	--	0.873	0.758
	EP2	0.812	12.461 ***		
	EP3	0.688	10.584 ***		
	EP4	0.731	11.191 ***		
	EP5	0.734	11.222 ***		
	EP6	0.698	10.840 ***		

Note: ‘a’ is set as a fixed value; *** *p <* 0.01.

**Table 3 ijerph-19-11787-t003:** Model Fit Indices for the Structural Model.

Model Fit Index	Value in the Study	Recommended Value
χ^2^/df	1.944	1.0 < χ^2^/df < 3.0
GFI	0.857	>0.90
AGFI	0.827	>0.90
RMR	0.027	<0.05
RMSEA	0.059	<0.08
PNFI	0.747	>0.50
PGFI	0.706	>0.50
CFI	0.914	>0.90
IFI	0.915	>0.90

**Table 4 ijerph-19-11787-t004:** Results of Hypothesis Testing.

Paths	Standardized Coefficient	t-Value	Result
Coercive pressures→Organizational learning	0.185 **	2.791	Supported
Mimetic pressures→Organizational learning	0.483 ***	6.327	Supported
Normative pressures→Organizational learning	−0.056	−0.892	N.S.
Coercive pressures→Environmental performance	0.261 ***	4.134	Supported
Mimetic pressures→Environmental performance	−0.054	−0.761	N.S.
Normative pressures→Environmental performance	0.045	0.788	N.S.
Organizational learning→Environmental performance	0.609 ***	6.976	Supported
Coercive pressures→Organizational learning→Environmental performance			Supported
Mimetic pressures→Organizational learning→Environmental performance			Supported
Normative pressures→Organizational learning→Environmental performance			N.S.

Note. *** *p* < 0.01, ** *p* < 0.05; N.S. denotes not supported.

**Table 5 ijerph-19-11787-t005:** Test the Mediating Role of Organizational Learning by Bootstrapping method.

Paths			Effect Type	Estimates	95% CI		Mediating Effect
Independent Variables	Dependent Variables	Mediating Variable			Lower Limit	Upper Limit	
Coercive pressure	Environmental performance	Organizational learning	Total effect	0.374	0.223	0.504	Partial
			Direct effect	0.261	0.128	0.385	
			Indirect effect	0.113	0.028	0.189	
Mimetic pressure			Total effect	0.240	0.112	0.387	Complete
			Direct effect	−0.054	−0.208	0.105	
			Indirect effect	0.294	0.199	0.413	
Normative pressure			Total effect	0.011	−0.100	0.128	None
			Direct effect	0.045	−0.046	0.150	
			Indirect effect	−0.034	−0.115	0.041	

## Data Availability

The datasets generated for this study are available on request to the corresponding author.

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
