# Peer review of "Is There a Missing Link? Exploring the Effects of Institutional Pressures on Environmental Performance in the Chinese Construction Industry"

_ijerph, 2022, doi:10.3390/ijerph191811787_

Round 1

Reviewer 1 Report

This paper investigates the mediating effect of organizational learning on institutional pressure and environmental performance by using survey data from construction industry enterprises in Shanxi Province, China. The research has theoretical value and practical significance. But in general, there are several issues that deserve deeper consideration:

1. In Hypothesis 1, it is unclear how institutional pressure acts on organizational learning. Environmental management practices should be the result of organizational learning, but from the existing content, the author seems to think that the two are the same thing. This leads to no essential difference between the contents of Hypothesis 1 and Hypothesis 2.

2. In Hypothesis 4, how does organizational learning mediate the relationship between institutional pressure and environmental performance? The argument needs to be strengthened.

3. Are there enough differences in institutional pressures faced by firms in the same province and industry to support the research content?

4. The contribution is insufficient and expression needs to be strengthened. The contribution of the first two points is the same. The influence and mechanism of institutional pressure on corporate environmental performance is not a new topic. What is the difference between this paper and the mechanism that has been revealed?

5. In line 23, “coercive pressures” should be changed to “mimetic pressures”. The reference format is not uniform, please check.

I hope the authors find these comments useful to improve their paper.

Author Response

Reviewer 1

Q1: In Hypothesis 1, it is unclear how institutional pressure acts on organizational learning. Environmental management practices should be the result of organizational learning, but from the existing content, the author seems to think that the two are the same thing. This leads to no essential difference between the contents of Hypothesis 1 and Hypothesis 2.

R1: We are grateful for the suggestion. As the reviewer suggested, we rephrase the contents of Section 2.2 to make a difference between Hypothesis 1 and Hypothesis 2.

Q2: In Hypothesis 4, how does organizational learning mediate the relationship between institutional pressure and environmental performance? The argument needs to be strengthened.

R2: We are grateful for the suggestion. As the reviewer suggested, we strengthen the argument on the mediation role of organizational learning in the relationship between institutional pressures and environmental performance (See Section 2.3.4).

Q3:Are there enough differences in institutional pressures faced by firms in the same province and industry to support the research content?

R3: Thanks for the question. Before we sent questionnaire to the managers of construction companies, we conducted interviews with some of the sample firms. Although firms are faced with the same institutional environment, we found that managers of the construction companies perceive differently with respect to the institutional pressures, which may lead to their different responses and therefore heterogeneous environmental performance.

Q4: The contribution is insufficient and expression needs to be strengthened. The contribution of the first two points is the same. The influence and mechanism of institutional pressure on corporate environmental performance is not a new topic. What is the difference between this paper and the mechanism that has been revealed?

R4: We are grateful for the suggestion. As the reviewer suggested, we rephrase the original first two points and strengthen the contribution of the paper.  

Previous studies have explored other mediators in the relationship between institutional pressures and environmental performance, such as training (Sarkis et al., 2010), knowledge resources (Simpson, 2012), comprehensiveness of environmental management systems (Phan & Baird, 2015), green organizational responses (Huang et al., 2016), organizational ambidexterity (Lin & Ho,2015), green supply chain management (GSCM) (Vanalle et al., 2017), and environmental concern (Juárez-Luis 2018). By uncovering the mediating mechanisms of organizational learning, we complement prior studies. The details are provided in Section 2.3.4.

Q5. In line 23, “coercive pressures” should be changed to “mimetic pressures”. The reference format is not uniform, please check.

R5: Thanks for the careful reviews. We check the papers recently published in IJERPH, and have the references in uniform format.

As the reviewer has requested, we rewritten the Introduction and Conclusion, and carefully screened the references to ensure relevance.

The cited references in the reply are included in the revised manuscript.

Reviewer 2 Report

This study argues that institutional pressures enable firms to go through organizational learning processes, affecting environmental performance. I agree that institutional theory is a valid theoretical lens to explain firm actions related to environmental issues. However, I found several points where the authors could consider for publication. Let me discuss the points below.

1.      Theory

The authors may want to trace the intellectual dialogues of institutional theory to theorize their main arguments. The isomorphism theses make sense, but readers may rarely expect more than isomorphism processes any more. There have been various discussions on isomorphism theoretically and empirically. Also, I don’t disagree that the organizational learning processes can be involved under institutional pressures. However, from the perspective of neo-institutionalism, organizational-learning processes are already contextualized. So I’m not sure how this paper contributes to the literature. The authors may want to clarify the theoretical contributions.

Furthermore, the authors may want to tighten their theoretical arguments along with their research design. I think the theory is not perfectly aligned to the research design, which is critical. I imagine that the authors may want to argue how project managers respond to the institutional pressures. These agentic responses to the institutional change can be the one this paper can take as the main perspective. However, as the isomorphic mechanisms were originally maintained in the organizational-field level, the theoretical frames may not fully match to the research design.

2.      Rationales

The rationales for the mediation process are not clear enough. Organizational learning is so broad that readers have their own interpretations rather than what the authors intend to do. Meanwhile, the authors seem to follow Baron & Kenny’s logic to examine mediation processes. And I found that the research design is about a company’s project managers’ perception processes upon the institutional pressures and their organizational responses to the perception. The authors may want to explain how and why project managers respond to certain institutional pressures and how the perception of an individual is disseminated to the organization entirely (i.e. organizational learning processes). Also, the authors may want to specify the decision-making processes to enhance environmental performance as a result of the learning processes. That is, under institutional pressures, how organization learns in what processes could be fully explained.

3.      Research design

Research design could be reconsidered. As discussed, the hypotheses are organization-level arguments, but their tests are made based on a certain manager’s perception. How the empirical approach can examine the causality the authors present could be fully elaborated. One possible remedy for this is to clearly state what is the research question of this study. The answers of the research question may be planned to be presented in the result section. To do it, the empirical approach could be re-designed.

Hope these comments help to develop the potential paper.

Author Response

Q1: The authors may want to trace the intellectual dialogues of institutional theory to theorize their main arguments. The isomorphism theses make sense, but readers may rarely expect more than isomorphism processes any more. There have been various discussions on isomorphism theoretically and empirically. Also, I don’t disagree that the organizational learning processes can be involved under institutional pressures. However, from the perspective of neo-institutionalism, organizational-learning processes are already contextualized. So I’m not sure how this paper contributes to the literature. The authors may want to clarify the theoretical contributions.

Furthermore, the authors may want to tighten their theoretical arguments along with their research design. I think the theory is not perfectly aligned to the research design, which is critical. I imagine that the authors may want to argue how project managers respond to the institutional pressures. These agentic responses to the institutional change can be the one this paper can take as the main perspective. However, as the isomorphic mechanisms were originally maintained in the organizational-field level, the theoretical frames may not fully match to the research design.

R1: We are grateful for the suggestions. We strengthen the contributions of this study in Section5.2. As the reviewer has noted, there have been various discussions on isomorphism theoretically and empirically. However, some authors have criticized and identified the limitations of the institution-theoretic framework (Lin & Ho, 2016; Huang and Chen, 2002; Hoffman, 2001). Institutional theory is weak in explaining why firms adopt changes whereas others do not, which leads to heterogeneity rather than homogeneity, even though they are embedded in the same institutional environment (Hoffman, 2001; Juárez-Luis et al., 2018). Thus, we aim to address the question of why firms facing common institutional pressures have heterogeneous environmental performance. In addition, the mixed and controversial findings on the relationship between institutional pressures and environmental performance may also suggest the mechanisms through which institutional pressures drive environmental performance is still largely unclear (Zhang & Zhu, 2019; Huang et al.,2016). Therefore, additional research is needed to explore the missing link between institutional pressures and environmental performance, which is one of the research gaps this study attempts to fill.

I agree with the reviewer that isomorphic mechanisms were originally maintained at the organizational-field level. In this study, institutional pressures of the organizational field—“those organizations that, in the aggregate, constitute a recognized area of institutional like: key suppliers, resources and product customers, regulatory agencies, and other organizations that produce similar services or products” (DiMaggio and Powell, 1983, 148). —are divided into three dimensions: coercive pressures from regulatory agencies, mimetic pressures from competitors, and normative pressures from suppliers and customers. We argue that firms’ responses in the form of active or reactive organizational learning to those institutional pressures affect their environmental performances. Following previous studies (e.g.,Phan & Baird,2015; Huang et al., 2016; Juárez-Luis et al. ,2018), we adopted the key informant technique  (Kumar et al., 1993) to select project managers in construction firms as the target respondents. Since the respondents were supposedly knowledgeable about the firms(s) and pressures from the industry stakeholders, the “perceptions” of the managers on the environmental practices actually denote the responses of the firms. Therefore, we test our hypotheses on the organization level.

Q2: The rationales for the mediation process are not clear enough. Organizational learning is so broad that readers have their own interpretations rather than what the authors intend to do. Meanwhile, the authors seem to follow Baron & Kenny’s logic to examine mediation processes. And I found that the research design is about a company’s project managers’ perception processes upon the institutional pressures and their organizational responses to the perception. The authors may want to explain how and why project managers respond to certain institutional pressures and how the perception of an individual is disseminated to the organization entirely (i.e. organizational learning processes). Also, the authors may want to specify the decision-making processes to enhance environmental performance as a result of the learning processes. That is, under institutional pressures, how organization learns in what processes could be fully explained.

R2: Thanks reviewer for the suggestions. As suggested by the reviewer, we clarify the definition of organizational learning in Section 2.2.  We also elaborate on the mediation mechanism of organizational learning in the relationship between institutional pressures and environmental performance in Section 2.3.4. My arguments are elaborated as follows. Based on Huang et al. (2016), the “pressure-response-performance” framework is adopted in this study. Specifically, Following Delmas & Toffel (2008), we use “organizational learning” to denote the organizational responses to external institutional pressures. In this study, we do not consider the learning process, rather we take an outcome-oriented view of organizational learning (Senge, 1990). As organizational learning is closely related to change and firms’ responses may be reactive or proactive, we further distinguish reactive and proactive organizational learning. While reactive organizational learning is associated with firms’ reactive posture to change and only making small adjustments to the existing environmental practices, proactive organizational learning reflects firms’ proactive posture to change and aims to switch to completely new forms of environmental practices. It is proposed that institutional pressures can and do lead to the improvement of a firm’s environmental performance in cases where a higher level of organizational learning is present. In other words, it is reactive or proactive organizational learning in response to institutional pressures that makes the difference in firms’ heterogeneous environmental performance.

Q3: Research design could be reconsidered. As discussed, the hypotheses are organization-level arguments, but their tests are made based on a certain manager’s perception. How the empirical approach can examine the causality the authors present could be fully elaborated. One possible remedy for this is to clearly state what is the research question of this study. The answers to the research question may be planned to be presented in the result section. To do it, the empirical approach could be re-designed.

R3: We are grateful and accept the reviewer’s suggestions. In the Introduction part, we clearly state the questions and objective of the study, and the answers to the research questions are presented in the result section. Drawing on organizational learning theory, we argue that a firm’s responses occur through managerial perceptions of its industry environment. In this study, managers of construction firms in central China were used as the key informants. Those managers possess the most comprehensive knowledge of the characteristics of the firm, its responses to external environment and firm performance. In short, the units of analysis were set as individual firms whereas the units of data collection were managers. In addition, similar studies (e.g., Simpson,2012,p36; Ye et al.,2013,p136; Juárez-Luis et al.,2018,p8 ) examining the influences of institutional pressures also use questionnaire are also based on managers’ perceptions, which shows the plausibility of our research design

As the reviewer has requested, we rewritten the Introduction and Conclusion, and carefully screened the references to ensure relevance.

The cited references in the reply are included in the revised manuscript.

Round 2

Reviewer 1 Report

The authors has responded to the concerns of review, and I have no furthur comments

Author Response

Reviewer 1 has no further comments.

Reviewer 2 Report

I appreciate for all the efforts for revising the paper and answering the questions I raised at the first round. However, I am not sure that the revised version can fully address what I was concerned about. First, the motivation of this paper could be re-framed. I am afraid that the context of construction industry cannot be a critical motivation for this study. The authors may want to directly talk about issues related to green washing or any window-dressing behaviors led by institutional pressures for environmental issues. I think starting with unexpected responses of firms to institutional pressures would be a more appealing and interesting motivation. Second, institutional theory should be further elaborated as a main theoretical framework. It is still unclear how institutional theory is employed to theoretically contribute to the literature. Third, the hypotheses are still clear enough. I expected more theoretically tightened hypotheses. In particular, I thought that the incorporation between institutional theory and organizational learning in terms of environmental performance can be one theoretical contribution, but in this current version, the link between institutional theory and organization learning is not clearly elaborated. Forth, the part of organizational learning is more problematic. Since organizational learning is a process or a mechanism, the concept should be specified according to the context. Yet, the arguments related to organizational learning is a bit broad so that it is not easy for me (as well as readers) to figure out how organizational learning engages in the link between institutional pressure and environmental performance. Last, the more critical issue on this paper is the measure of organizational learning. As discussed, since organizational learning is a process, it should be contextualized. That is, the measure for organizational learning should be capture any actions taken for environmental issues. Given that the measure in this study is a general measure on learning in an organization, I am afraid that it cannot actually test the hypotheses.

Hope these comments are helpful to develop this paper.

Author Response

I really appreciate your comments for improving the paper in Round 2.

Based on the instructions, we uploaded the file of the revised manuscript with all the changes highlighted by using the track changes mode in MS word.

Appended to this letter is our point-by-point response to the comments. The comments are reproduced and our responses are given directly afterward in a different color (red).

We would like also to thank you for allowing us to resubmit a revised copy of the manuscript.

Here are the general comments from the reviewer 2.

I appreciate for all the efforts for revising the paper and answering the questions I raised at the first round. However, I am not sure that the revised version can fully address what I was concerned about.

Q1: First, the motivation of this paper could be re-framed. I am afraid that the context of construction industry cannot be a critical motivation for this study. The authors may want to directly talk about issues related to green washing or any window-dressing behaviors led by institutional pressures for environmental issues. I think starting with unexpected responses of firms to institutional pressures would be a more appealing and interesting motivation.

R1: Thank you for your suggestion. We do think that it is a better choice by discussing the green-washing or window-dressing behavior of firms under institutional pressures. We rephrase the Section Introduction (See Page 2, Line 50 to Line 66).

Q2: Second, institutional theory should be further elaborated as a main theoretical framework. It is still unclear how institutional theory is employed to theoretically contribute to the literature.

R2: Thank you for your suggestion. We further elaborate the institutional theory, especially the limitations of the theory, see Page5, Line 222 to Line 233.

Q3: Third, the hypotheses are still clear enough. I expected more theoretically tightened hypotheses. In particular, I thought that the incorporation between institutional theory and organizational learning in terms of environmental performance can be one theoretical contribution, but in this current version, the link between institutional theory and organization learning is not clearly elaborated.

R3: Thank you for your suggestion. Considering the limitations of institutional theory described in Section 2.1, both institutional and organizational characteristics are needed to be jointly considered to understand the heterogeneity in performance [12,21]. An extension of the institutional theory in this study therefore is the proposition of organizational learning as a mediator in the relationship between institutional pressures and environmental performance. This point is elaborated as follows. First, organizational learning is essentially about adapting and responding to changes in the institutional environment and is one of the direct approaches for firms to understand and respond to institutional pressures [12,48]. In other words, an organization learns to develop organizational structures and systems to transform itself to become more adaptive and responsive to changes and jolts in the external environment[49]. As Meyer [50] argued, such environmental jolts provide a good opportunity for an organization to learn to deal with external environmental pressures. For example, in response to progressive health and safety legislative requirements, construction firms have begun to learn and apply for relevant certifications, such as environmental management systems and Occupational health and safety management systems (ISO45001). Second, organizational learning is gradually developed in the continuous interaction and coordination of various explorative and exploitative actives. This dynamic capability focuses on continuous adaptation to the changing external environment. In addition, organizational learning is regarded as a means to foster sustainability by institutionalizing new thinking [43]. Thus, “organizational learning” is used in this study to denote the organizational responses to external institutional pressures. It is the different levels of organizational learning which lead to the differences in environmental performance. As such, by linking institutional theory and organizational learning theory, we may provide explanations on questions of why firms facing common institutional pressures have heterogeneous environmental performance [12,16,20].

Q4: Forth, the part of organizational learning is more problematic. Since organizational learning is a process or a mechanism, the concept should be specified according to the context. Yet, the arguments related to organizational learning is a bit broad so that it is not easy for me (as well as readers) to figure out how organizational learning engages in the link between institutional pressure and environmental performance.

R4: Thank you for your suggestion. As you suggested, the concept of organizational learning and the associated hypotheses are specified to the context of the construction industry.

Regarding the concept of organizational learning, according to Bell et al. (2002), there exist four disparate schools of thought on organizational learning: an economic school, a developmental school, a process school, and a managerial school. Among them, the managerial school argued that organizational learning takes place as a direct result of management-led intervention. Managerial intervention is frequently a carefully calculated corporate response to perceived environmental pressures. Because the external environment is constantly changing, a firm must also make continuous increases in its “capacity to realize [its] highest aspirations” [40]. This definition emphasizes a firm’s behavior changes and capacity for action, which is applied in this study. Specifically, considering the context construction industry, we describe organizational learning as the way of construction firms act (reactive vs. proactive) and the extent of changes in practices related to environmental management in response to external environmental pressures. Those environmental management practices include the techniques and practices employed by an organization to tackle environmental challenges, such as building information modeling (BIM), post occupancy evaluation (POE), total quality management (TQM), just-in-time (JIT), and environmental management systems(EMS) [13-15,41]. The most intuitive feature of BIM is the 3D visualization in the early stage, which can optimize the engineering design, and reduce the possibility of material loss and rework that may exist in the construction stage. The cyclical aspect of continuous improvement of POE, TQM, JIT, and EMS also has as core tenets the concept of waste reduction and elimination, even so far as going to a ”zero waste” goal [13]. Such techniques and policies, aimed at reducing, reusing, and recycling waste, are helpful for construction firms to improve environmental performance. As organizational learning is closely related to (planned) changes of organizations and firms’ responses may be reactive or proactive, the study distinguishes between reactive and proactive organizational learning. While reactive organizational learning entails small adjustments or adaptations of the ways the organizations act (e.g. adjusting existing routines, actions, products, and processes), proactive organizational learning requires far-reaching changes, which can also aim at a switch to completely new forms of satisfying customer needs (e.g., green building technology and process innovations) [42]. Extreme situation of “zero learning” exists when firms were unaware of all of the potential waste reduction options [30], or ignored the implementation of green practices [43].

Q5: Last, the more critical issue on this paper is the measure of organizational learning. As discussed, since organizational learning is a process, it should be contextualized. That is, the measure for organizational learning should be capture any actions taken for environmental issues. Given that the measure in this study is a general measure on learning in an organization, I am afraid that it cannot actually test the hypotheses.

R5: we contextualize the measures for organizational learning in the context of construction industry. We enclosed an appendix of measurement items for each construct at the end of the paper. The measurement items for organizational learning are as follows, such as “Our company proactively searches for advanced construction technologies (e.g., assembled building technology, BIM) that would reduce waste.”, “Our company proactively promotes the implementation of clean production processes and technologies in construction projects.”,” Our company proactively searches for information to refine common methods and ideas in solving problems in construction projects.”, “Our company changes the existing high energy-consumption technology and switches to energy-saving and environmental protection technology.”, “Our company proactively learns through feedback from previous construction projects, rather than merely reacting to the detected issues.”.

As the reviewer has requested, we carefully screened the references to ensure relevance, with some references added and some deleted.

The cited references in the reply are included in the revised manuscript.
